# Prediction of Spatiotemporal Invasive Risk of the Red Import Fire Ant, *Solenopsis invicta* (Hymenoptera: Formicidae), in China

**DOI:** 10.3390/insects12100874

**Published:** 2021-09-27

**Authors:** Jinyue Song, Hua Zhang, Ming Li, Wuhong Han, Yuxin Yin, Jinping Lei

**Affiliations:** 1College of Geography and Environment Science, Northwest Normal University, Lanzhou 730070, China; 2019212355@nwnu.edu.cn (M.L.); 2019212364@nwnu.edu.cn (W.H.); 2020212680@nwnu.edu.cn (Y.Y.); 2020212672@nwnu.edu.cn (J.L.); 2Key Laboratory of Resource Environment and Sustainable Development of Oasis, Lanzhou 730070, China

**Keywords:** *S. invicta*, MaxEnt, suitable area, environmental factors

## Abstract

**Simple Summary:**

The red imported fire ant, *Solenopsis invicta* (Hymenoptera: Formicidae), is an invasive pest, and it has spread rapidly all over the world. In this study, based on the distribution data and environmental factor data of *S. invicta*, the optimized MaxEnt model was used to predict the suitable areas of *S. invicta* growth in China under current and future climatic conditions. The results show that the potential suitable area of *S. invicta* growth in the current climate is 81.37 × 10^4^ km^2^ in size and is mainly located in the south and southeast of China. The total suitable area of *S. invicta* growth is expected to increase in the future climate change scenario, and the suitable area is likely to spread to higher latitudes.

**Abstract:**

The red imported fire ant, *Solenopsis invicta* (Hymenoptera: Formicidae)*,* is an invasive pest, and it has spread rapidly all over the world. Predicting the suitable area of *S. invicta* growth in China will provide a reference that will allow for its invasion to be curbed. In this study, based on the 354 geographical distribution records of *S. invicta*, combined with 24 environmental factors, the suitable areas of *S. invicta* growth in China under current (2000s) and future (2030s and 2050s) climate scenarios (SSPs1-2.5s, SSPs2-3.5s and SSPs5-8.5s) were predicted by using the optimized MaxEnt model and geo-detector model. An iterative algorithm and knife-cut test were used to evaluate the important environmental factors that restrict the suitable area under the current climatic conditions. This study also used the response curve to determine the appropriate value of environmental factors to further predict the change and the center of gravity transfer of the suitable area under climate change. The optimized MaxEnt model has high prediction accuracy, and the working curve area (AUC) of the subjects is 0.974. Under climatic conditions, the suitable area is 81.37 × 10^4^ km^2^ in size and is mainly located in the south and southeast of China. The main environmental factors affecting the suitable area are temperature (Bio1, Bio6, and Bio9), precipitation (Bio12 and Bio14) and NDVI. In future climate change scenarios, the total suitable area will spread to higher latitudes. This distribution will provide an important theoretical basis for relevant departments to rapidly prevent and control the invasion of *S. invicta*.

## 1. Introduction

Biological invasion has become one of the global environmental problems of the 21st century [1]. With the intensification of the process of global economic integration, China’s trade and transportation industry is developing rapidly, biological invasion occurs increasingly frequently, and the harm of invasive species is becoming increasingly serious [2]. Since the 20th century, large-scale biological invasions have occurred mainly in farmland, forests, grasslands, islands, wetlands, rivers, oceans, and nature reserves [3].

In 2012, relevant scholars compiled a list of alien invasive species in China, listing a total of 488 invasive species living in terrestrial ecosystems, inland waters, and marine ecosystems, including 171 species of animals, 265 species of plants, 26 species of fungi, 3 species of protozoa, 11 species of prokaryotes, and 12 species of viruses [4]. By the end of 2016, there were at least 610 alien invasive species in China, of which 50 were listed as the 100 most harmful alien invasive species (ICUN) in the world [2]. For biological invasion, prevention is more critical than governance [5].

The prevention and control of alien species has become a top priority in many places. Climatic and environmental factors are not only key factors affecting the number of populations and physiological metabolisms, but they are essential in determining the distribution range of species [6]. The National Center for Atmospheric Research demonstrated that, with the increase in greenhouse gas emissions and the destruction of the ozone layer, the earth surface temperature shows an upward trend [7], with the temperature in China increasing by 1.6–5.0 °C [8].

The successful invasion of exotic pests will not only reduce the biodiversity and ecosystem services of the invading areas but also cause huge losses to the global economy, environment and human health [9,10]. Therefore, it is of great practical significance to study the potential geographical distribution of invasive species in the context of future climate change for the monitoring and prevention of their occurrence areas and the early identification and supervision of non-occurrence areas.

*S. invicta* (Hymenoptera: Formicidae) is a social insect. It is native to South America. In 2003, it was discovered in Taoyuan and Chiayi, Taiwan [11]; it invaded Guangdong [12], China in 2004 and then rapidly spread to 435 counties and districts in 12 provinces, causing serious harm [13]. The loss caused by *S. invicta* invasion in China was calculated to total USD 25 billion [14]. *S. invicta* is listed by the Ministry of Ecology and Environment of the People’s Republic of China as the second batch of alien invasive species in China and by the World Conservation Union (IUCN) as one of the 100 most destructive species in the world due to their miscellaneous food habits [12], strong competitiveness, and obvious harm to the natural environment, human health, agriculture, and forestry.

At present, foreign research mainly focuses on the geographical distribution and restrictive factors of *S. invicta* in two ways: the population dynamics model based on biological characteristics and the biological distribution hypothesis based on climate constraints. Korzukhin et al. discussed, in detail, the northern boundary of the distribution of *S. invicta* in the United States by using the population dynamics model and a large number of survey data from various states [15,16].

Morrison et al. estimated the distribution of *S. invicta* around the world [17]. Sutherst et al. analyzed the adaptability of the United States and the world by using the Climex model, focusing on the impact of regional climatic factors on the distribution of *S. invicta* [18]. As there are few studies on the distribution of *S. invicta* in China using MaxEnt model, it is deemed of upmost importance to use the optimized version of it to predict the suitable growth zone and migration route of *S. invicta* in current and future climatic conditions in China.

The optimized MaxEnt model was used to predict the suitable area of *S. invicta* growth in China under current and future climatic conditions. The main goals of the current study are as follows: (1) screening out the main environmental factors by combining the factor detector in the geo-detector model and the knife-cutting method in the MaxEnt model; (2) using ENMeval to screen the optimal parameters of the model; and (3) predicting the suitable area, the change, and the migration route of *S. invicta* under current (2000) and future (2030, 2050) climatic conditions. The results provide a scientific basis for the effective prevention of the invasion of *S. invicta* to a wider area.

## 2. Materials and Methods

### 2.1. Species Overview

*S. invicta* is a social insect that can spread in many ways, mainly natural migration, spread with floods, and goods circulation. *S. invicta* are extremely harmful, and their main characteristics are strong aggression and high bite toxicity. They are very aggressive to people, with an average of 438 out of 100,000 people in the invading areas being attacked [19]. After being bitten by invasive *S. invicta*, the skin will display erythema, swelling, pain, itching, thickening, and deformity, and it can even result in high fever, pain, shock, and death in serious cases, seriously endangering human health [20]. In addition, the pest also causes serious damage to crops, forests, other animals, and public facilities in rural or urban areas [21].

The most common nests of *S. invicta* are mounds near the green belts and the roots of trees. Temperature and rainfall in climate variables play an important role in determining the distribution of *S. invicta* [22]. In terms of temperature, the lowest temperature of heat tolerance of *S. invicta* is 3.6 °C, and the highest temperature is 40.7 °C. *S. invicta* begin to look for food when the soil surface temperature is above 10 °C and do not stop foraging until the soil temperature reaches 19 °C. The range of soil surface temperature for foraging is 12–51 °C [23]. In terms of rainfall, continents with more annual rainfall than 510 mm can support the survival of *S. invicta*, while, on continents with less annual rainfall than 510 mm, they may live close to permanent water sources (such as lakes, rivers, and springs) or regularly irrigated areas (such as fields and lawns) [17,24].

### 2.2. Data and Processing of Species Distribution

The MaxEnt model needs species distribution data and environmental data. The distribution data of *S. invicta* was obtained by the following: (1) search for periodical papers and relevant government reports with “*S. invicta*“ as the key words; (2) query about the species distribution databases (the databases queried in this study include the International Center for Agriculture and Biosciences (CABI, http://www.cabi.org accessed on 4/April/2021), the GBIF (http://www.gbif.org accessed on 5/April/2021), and the National Pest Quarantine Information platform of China (http://www.pestchina.com accessed on 6/April/2021); and (3) referral to the National list of Agricultural Plant Quarantine pests Distribution (http://www.moa.gov.cn/ accessed on 6/April/2021) published by the Ministry of Agriculture in 2012–2019 and forestry pests in China [25].

Data processing: The buffer module in the neighborhood analysis of ArcGIS software was used to eliminate the distributed data points to ensure that each grid contains only one distribution point so as to avoid over-fitting caused by too many distribution points. In this study, the spatial resolution of environmental data was 2.5 arc-minutes (about 4.5 km^2^), and the buffer diameter was 3 km. When the distance between the two distribution points was less than 3 km, only one of the distribution points was retained [26]. The longitude and latitude of each distribution point were determined by Google, and a total of 354 distribution points were obtained (Figure 1).

### 2.3. Environmental Factors and Processing

The climate data used in this study (1970–2000) were obtained from the World Climate Data website (http://www.worldclim.org/, accessed on 7/April/2021). Future climate data (2021–2040 and 2041–2060) were based on the BCC-CSM2-MR climate system model developed by the National Climate Center [27]. The model includes four estimated emission scenarios in SSPs proposed by the 6th International Coupled Modes Comparison Plan (CMIP6) [28]. The data include 24 variables with a data resolution of 2.5′ (Table 1).

As there is a certain correlation between the environmental factors, the geo-detector model (http://www.geodetector.org/ accessed on 10/April/2021) was used to eliminate the environmental factors with small contributions. This research uses the factor detector in the geo-detector model [29], which can detect the extent to which environmental factors affect the suitable area of *S. invicta* growth and obtain the influence value (*q* value) and factor explanatory value (*p* value) of each factor. The current suitable area of *S. invicta* growth and the environmental factor values were matched, and factor detection analysis was carried out to obtain the q value. The environmental factors had a q value greater than 0.1. Finally, nine environmental factors were selected for follow-up modeling (Table 1).

### 2.4. MaxEnt Model Construction and Parameter Optimization

Ecological niche modeling uses the known distribution data, relevant environmental variables of species, and the ecological needs of species, and it projects the results to different times and spaces to predict the potential distribution of species. Common ecological niche models include BIOCLIM, DOMAIN, GARP, CLIMEX, and MaxEnt [30]. Compared with other ecological niche models, the MaxEnt model (http://biodiversity/informatics.amnh.org/open/source/Maxent, version 3.4.1, accessed on 12/April/2021) is less affected by sample deviation [31], strong stability, and high accuracy and has the ability to predict habitat suitability of alien invasive organisms [32,33,34].

When the default parameters of the MaxEnt model are used to simulate the potential distribution of species, over-fitting can easily occur, and the prediction results may not reflect the actual situation [31]. This problem can be avoided by the adjusting function feature combination (feature combination, FC) and regulated frequency doubling (regularization multiplier, RM). In practice, when using the MaxEnt model to simulate species distribution, most researchers use default parameters. The default parameters of MaxEnt are derived from 226 species [35].

Studying the species distribution under the default parameters, over-fitting and high complexity can occur, and this may also reduce the accuracy of the research results; therefore, it is necessary to optimize the model. The AICc index is a standard tool used to measure the goodness of statistical model fitting, which can avoid over-fitting of data. ENMeval data packets can be used in order to adjust the parameters of MaxEnt model regulation frequency doubling (RM) and feature combination (FC). The minimum value of AICc was selected as the optimal setting, and the final model was established [36]. The *S. invicta* fitness values predicted by MaxEnt model are continuous raster data; the values are between 0 and 1; and the closer the values are to 1, the higher the possibility of species existence. The grades of suitable area are as follows: unsuitable area < 0.1; 0.1 ≤ poorly suitable area < 0.3; 0.3 ≤ moderately suitable area < 0.6; and highly suitable area ≥ 0.6.

### 2.5. MaxEnt Model Evaluation

At present, the most widely used method to evaluate the accuracy of the model is the ROC curve method (AUC method). As AUC is not affected by the diagnostic threshold and can provide performance evaluation results on all threshold ranges, it is recognized as the best evaluation index in the field of niche model evaluation [37]. The range of the AUC value is 0–1, and the simulation prediction grades are very bad (AUC ≤ 0.80), good (0.80 < AUC ≤ 0.90), better (0.90 < AUC < 0.95), and excellent (0.95 < AUC < 1.00) [38].

## 3. Results

### 3.1. Optimal Model and Accuracy Evaluation

Based on the number of species distribution points of *S. invicta*, the MaxEnt model RM = 1 (Figure 2). The mean AUC value was higher when the parameter settings were FC = LQH and RM = 3, indicating the optimized model’s prediction of *S. invicta*. However, when the parameters were set to FC = LQH and RM = 3, the values of mean AUC.diff and mean OR10 were significantly lower than those of the Maxent model under the default parameters (Figure 2), indicating that the optimized model is helpful in reducing the complexity of the model, avoiding over-fitting of species data, and enhancing the ability to predict the suitable area.

The MaxEnt model was reconstructed based on the optimal combination, and the ROC curve is shown in Figure 3, in which the training AUC is 0.974 and the test AUC is 0.973.

### 3.2. Dominant Environmental Factors

The contribution of the NDVI was the highest, the average temperature contribution rate of the driest season was 20.4%, the driest monthly precipitation contribution was 18.5%, the min temperature of the contribution of the coldest month was 14.6%, the annual precipitation contribution was 14.2%, the annual average temperature contribution was 7.9%, and the annual temperature range contribution was 1.1%. The contribution of altitude and land use were the lowest.

According to the knife-cutting method in the MaxEnt model, only considering the use of a single environmental variable, the six environmental variables that have the greatest influence on the regularization training gain are as follows: annual average temperature, min temperature of coldest month, mean temperature of driest quarter, temperature annual range, annual precipitation, and coefficient of variation of precipitation seasonality (Figure 4). Therefore, in this study, the environmental variables with a contribution rate of more than 10% and the first three environmental variables affected by positive planning training gain were selected as the main environmental variables affecting the distribution of *S. invicta*, which were the annual mean temperature, minimum temperature of the coldest month, mean temperature of the driest quarter, annual precipitation, coefficient of variation of the precipitation seasonality, and NDVI.

The running results of MaxEnt model show that when the probability of the presence of *S. invicta* is ≥0.6 and the grade of the suitable area is high, the NDVI ranges from 0 to 0.58. When the NDVI is 0.33, the probability of the presence of *S. invicta* reaches the maximum, and the mean temperature of the driest quarter is −13.18 to 22.68 °C. When the mean temperature of the driest quarter is 22.68 °C, the probability of the presence of *S. invicta* reaches the maximum. The seasonal variation coefficient of precipitation ranges from 0.11 to 133.21, and when the seasonal variation coefficient of precipitation is 133.21, the probability of the presence of *S. invicta* reaches the maximum and the min temperature of the coldest month is 4.21–5.52 °C (Figure 5).

When min temperature of coldest month is 5.52 °C, the probability of the presence of *S. invicta* reaches the maximum. The range of annual rainfall is 15.36–3092.69 mm, and when the annual rainfall is 872.98 mm, the probability of the presence of *S. invicta* reaches the maximum, and the annual average temperature ranges from 16.75–25.75 °C. Finally, the probability of the presence of *S. invicta* reaches the maximum value when the average temperature of the year is 25.75 °C.

### 3.3. Suitable Areas under Current Conditions

Under the current climatic conditions, the suitable areas of *S. invicta* growth are mainly concentrated in Yunnan, Guangxi, Guangdong, Hainan, Fujian, eastern Sichuan, western Chongqing, and southern Jiangsu. Highly suitable areas are southern Guangdong, coastal areas of Fujian, western Taiwan, Guangxi Province, and sporadic areas of Yunnan Province, which are in eastern Yunnan Province and Guangxi and Guangdong Province, Hunan, Jiangxi, Zhejiang, and Fujian Province (Figure 6). The total suitable area of *S. invicta* growth under the current climatic conditions is 81.37 × 10^4^ km^2^ in size, accounting for 8.48% of the total land area. The highly, moderately, and poorly suitable areas account for 3.98%, 13.10%, and 82.92% of the total suitable area, respectively (Table 2).

### 3.4. The Suitable Areas under Future Climate Change Scenarios

Compared with the prediction results of the suitable area of *S. invicta* growth in China under the current climatic conditions, the suitable area of *S. invicta* growth is increasing (Figure 7). The main characteristics are as follows:

(1) The SSPs1-2.6 scenario: The total suitable area of *S. invicta* growth increased by 21.64% and 39.14%, respectively in 2030 and 2050; the highly suitable areas of *S. invicta* growth increased by 19.44% and 32.41% in 2030 and 2050, respectively; the moderately suitable areas of *S. invicta* growth increased by 35.46% and 53.00% in 2030 and 2050, respectively; the poorly suitable areas of *S. invicta* growth increased by 25.30% and 37.28% in 2030 and 2050, respectively.

(2) The SSPs2-4.5 scenario: the total suitable area of *S. invicta* growth increased by 32.95% and 58.68% in 2030 and 2050, respectively; the highly suitable areas of *S. invicta* growth increased by 23.46% and 54.94% in 2030 and 2050, respectively; the moderately suitable areas of *S. invicta* growth increased by 44.47% and 82.18% in 2030 and 2050, respectively; the poorly suitable areas of *S. invicta* growth increased by 31.58% and 55.15% in 2030 and 2050, respectively.

(3) The SSPs5-8.5 scenario: the total area of suitable area of *S. invicta* growth increased by 39.04% and 68.01% in 2030 and 2050, respectively; the highly suitable area of *S. invicta* growth increased by 31.48% and 68.52% in 2030 and 2050, respectively; the moderately suitable area of *S. invicta* growth increased by 50.56% and 96.44% in 2030 and 2050, respectively; the poorly suitable areas of *S. invicta* growth increased by 37.59% and 63.50% in 2030 and 2050, respectively.

The distribution center of *S. invicta* under the current climatic conditions was located in Liuzhou City, Guangxi Province. As can be seen from Table 3 and Figure 8 and Figure 9, the main characteristics are given below:

(1) The SSPs1-2.6 scenario: The distribution center latitude increased by about 0.47°, the suitable area increased by 19.32 × 10^4^ km^2^, the decreased area was 0.08 × 10^4^ km^2^, and the maximum proportion of stable area was 72.96 × 10^4^ km^2^. The suitable areas mainly appear in northern Guangxi, northern Guangdong, and eastern Sichuan Province. Yunnan, Guizhou, Hunan, Jiangxi, and Fujian Province have fragmentary distribution. Hubei, Anhui, Zhejiang, Jiangsu, Henan, southern Tibet Autonomous Region, southern Shandong, and southern Shaanxi Province all displayed a small distribution.

The latitude of the *S. invicta* distribution center in 2050 increased by about 0.62°, the suitable area increased by 28.42 × 10^4^ km^2^, the decreased area was 0.14 × 10^4^ km^2^, and the stable area was 72.89 × 10^4^ km^2^. The distribution of suitable areas was larger than that in 2030, mainly in northern Guangxi, northern Guangdong, and eastern Sichuan, and fragmented distribution was observed in Yunnan, Guizhou, Hunan, Jiangxi, and Fujian. Hubei, Anhui, Zhejiang, Jiangsu, Henan, central Taiwan, southern Tibet Autonomous Region, southern Shandong, and southern Shaanxi have displayed a small amount of sporadic distribution.

(2) The SSPs2-3.5 scenario: The latitude of the *S. invicta* distribution center in 2030 increased by 0.59°, the suitable area increased by 23.89 × 10^4^ km^2^, the decreased area was 0.04 × 10^4^ km^2^, and the suitable area was 73.00 × 10^4^ km^2^. The new suitable areas mainly appear in northern Guangxi, northern Guangdong, and eastern Sichuan Province. Yunnan, Guizhou, Hunan, Jiangxi, and Fujian Province have fragmentary distribution. Hubei, Anhui, Zhejiang, Jiangsu, Henan, central Taiwan, southern Tibet Autonomous Region, and southern Shaanxi Province all displayed a small distribution.

The latitude of the *S. invicta* distribution center in 2050 increased by 1.09°, the suitable area increased by 42.91 × 10^4^ km^2^, the decreased area was 0.11 × 10^4^ km^2^, and the proportion of the stable area was 72.93 × 10^4^ km^2^. The suitable areas were mainly in northern Guangxi, northern Guangdong, northern Fujian, eastern Sichuan, eastern Hunan, and southern Hubei Province. Fragmentary distribution was found in Tibet Autonomous Region, Yunnan, Guizhou, Jiangxi, Zhejiang, Anhui, Jiangsu, and other provinces, while small distribution was observed in Henan, Shandong, and Shaanxi.

(3) The SSPs5-8.5 scenario: The latitude of the *S. invicta* distribution center in 2030 increased by 0.80°, the suitable area increased by 28.60 × 10^4^ km^2^, the decreased area was 0.05 × 10^4^ km^2^, and the suitable area was 72.99 × 10^4^ km^2^. The suitable areas were mainly in northern Guangxi, northern Guangdong, and eastern Sichuan. Yunnan, Guizhou, Hunan, Jiangxi, and Fujian Province displayed fragmentary distribution. Hubei, Anhui, Zhejiang, Jiangsu, Henan, southern Tibet Autonomous Region, southern Shandong, and southern Shaanxi Province all displayed a small distribution.

The latitude of the *S. invicta* distribution center in 2050 increased by 1.12° compared with the current distribution center, the suitable area increased by 49.44 × 10^4^ km^2^, the decreased area was 0.01 × 10^4^ km^2^, and the suitable area was 73.03 × 10^4^ km^2^. The suitable areas were mainly in northern Guangxi, northern Guangdong, eastern Sichuan, Guizhou, Hunan, Hubei, Jiangxi, Fujian, and other provinces. Tibet Autonomous Region, Yunnan, Zhejiang, Jiangsu, Anhui, and other provinces displayed fragmented distribution, while southern Shaanxi, Shandong, and Henan provinces displayed a small amount of sporadic distribution.

## 4. Discussion

In this study, the optimized MaxEnt niche model was used to predict the suitable areas of *S. invicta* growth in China under current and future climatic conditions. We found that, under the current climatic conditions, the range of suitable areas of *S. invicta* growth was 18.15–35.04° N, 91.66°–122.03° E, and the northern boundary reached southern Tibet, northern Yunnan, eastern Sichuan, central Chongqing, southern Hubei, southern Henan, central Anhui, and southern Jiangsu. The center of gravity of the geometric distribution of *S. invicta* was generally predicted in this study, and the range of distribution of suitable areas gradually spread to the northeast, suitable areas in Hunan, Hubei, Jiangxi, Zhejiang, Jiangsu, Shandong, and other provinces.

In the future climate change scenario, the increased area of *S. invicta* growth was much larger than that of the loss, indicating that *S. invicta* will continue to spread. Chen et al. take the days higher than the threshold temperature of the development of *S. invicta* and the effective accumulated temperature as decisive factors and the average annual precipitation and altitude as limiting factors [39]. It was predicted that most of Guangdong, central and southern Guangxi, southern Yunnan, Hainan, Taiwan, Hong Kong, and Macao are the main suitable areas of *S. invicta* growth.

Shen Wenjun used a similar deviation method to predict the suitable areas of *S. invicta* growth [40]. The prediction results show that Hong Kong, Macao, Guangdong, Guangxi, Anhui, Jiangsu, Shanghai, Jiangxi, Zhejiang, Fujian, Taiwan, Guizhou, Yunnan, Hainan, and other regions have a high probability of invasion. In terms of the overall trend, these two methods and the methods in this paper had similar results.

Therefore, this study optimized the parameter setting of the MaxEnt model based on AICc information criterion, which can effectively avoid overfitting and better predict species distribution in line with the Shelford tolerance rule (law of tolerance) [41]. The AUC value tested by the model is 0.974, which shows that the prediction result of the optimized model has high accuracy and credibility, and the prediction result is consistent with the actual distribution of *S. invicta* in China.

Among many environmental factors, we selected nine climatic, topography, and land use factors to predict the suitable area of *S. invicta* growth. According to the prediction results of the MaxEnt model, the main environmental factors affecting the distribution of *S. invicta* are annual average temperature, lowest temperature in the coldest month, average temperature in the driest season, annual rainfall, seasonal coefficient of variation of precipitation, and the NDVI.

This study shows that the probability of the presence of *S. invicta* increases slowly when the NDVI value is 0–0.58, while when the value is between 0.58 and 1.0, its probability of the presence begins to decrease, and with the increase in the lowest temperature in the coldest month, the average temperature in the driest season, and the seasonal variation of precipitation, the probability of the presence of *S. invicta* also increases. When the annual rainfall is 15.36–872.98 mm, the probability of the presence of *S. invicta* increases slowly, while when the rainfall is 872.98–3092.69 mm, the probability of the presence of *S. invicta* decreases, and when the average temperature reaches 1.25 °C, the probability of the presence of *S. invicta* increases.

Lei studied the optimum activity temperature of *S. invicta* and found that the optimum activity temperature was 26 °C, followed by 22 °C and 18 °C [42]. Sutherest et al. used 17 °C, 21 °C, and 24 °C as the starting temperatures to simulate the growth of *S. invicta*. It was found that the simulation results of 17 °C were consistent with the development of larvae and pupae and the actual activity of the population. In this study, the best range of annual average temperature of *S. invicta* was 16.75–25.75 °C, and the probability of the presence of *S. invicta* reached the maximum when the annual average temperature is 25.75 °C, which proves that the results of this study are credible.

For invasive species, in addition to being affected by climatic conditions, the distribution may also be related to the spread speed of red fire ants, which spread outward at a rate of 198 km per year in the southern area of the United States [43]. The reason why red fire ants spread fast is that they have a strong reproductive ability, and a queen can produce about 7000 workers [44]. In the actual living environment of the species, interspecific competition, species propagule pressure, human interference, and climate change all have an impact on predicting its potential distribution [45,46]. Therefore, the suitable areas of *S. invicta* predicted in this study should be combined with the actual situation of the invaded areas, and then the formation of relevant prevention plans and control measures should be carried out.

Combined with the results of this study, in order to prevent the invasion and spread of *S. invicta*, according to the occurrence law and biological characteristics of *S. invicta*, monitoring points should be set up in suitable areas, highly suitable areas, and areas with suitable climatic conditions for the growth and reproduction of *S. invicta*. We should take positive and effective measures to prevent and control their harm, aim to reduce the economic losses caused by disasters, and effectively slow down the speed of this species’ outward spread.

## 5. Conclusions

The results show that invasive *S. invicta* can survive in at least 12 provinces of mainland China, especially in South China with high temperature, humidity, and rainfall. Among the three climate change scenarios and the total suitable area of *S. invicta* growth, the increase in the highly suitable area of *S. invicta* growth is much larger than that in the lost area, indicating that *S. invicta* may not face extinction and will continue to spread in the future. Therefore, *S. invicta* occurrence areas should be monitored and controlled in addition to early identification and supervision of non-occurrence areas.

## Figures and Tables

**Figure 1 insects-12-00874-f001:**
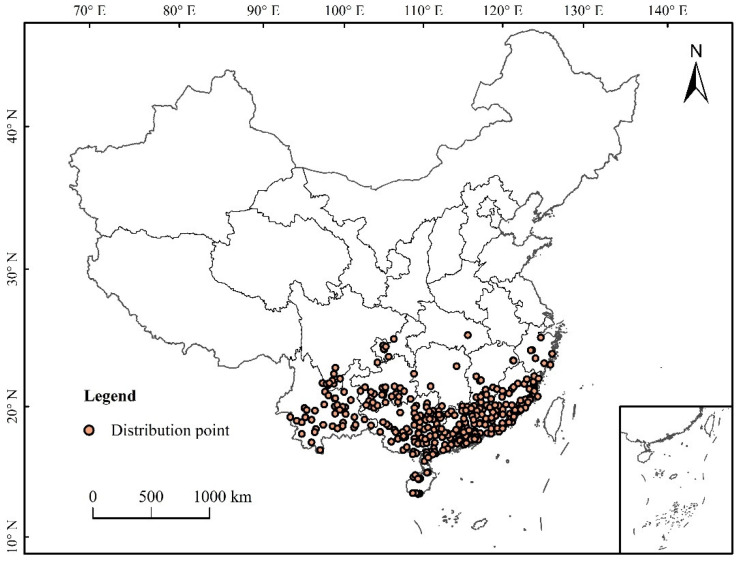
Distribution of *S. invicta * in China.

**Figure 2 insects-12-00874-f002:**
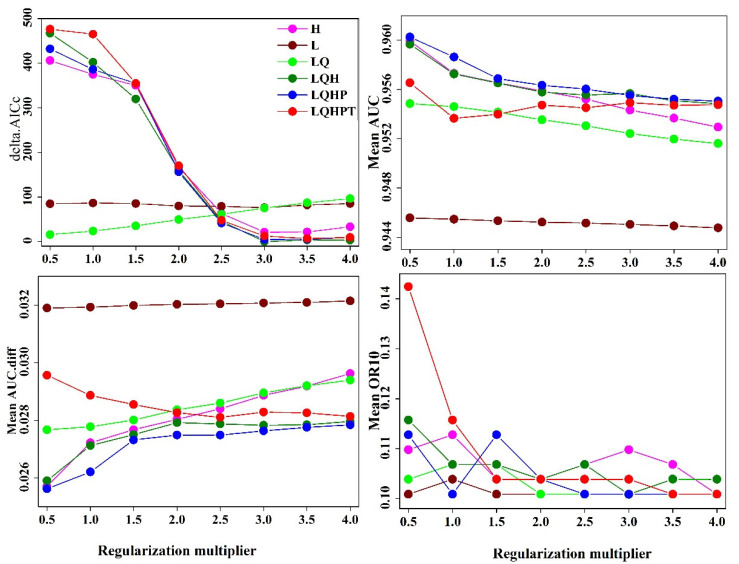
Evaluation results of MaxEnt model *S. invicta * under different settings.

**Figure 3 insects-12-00874-f003:**
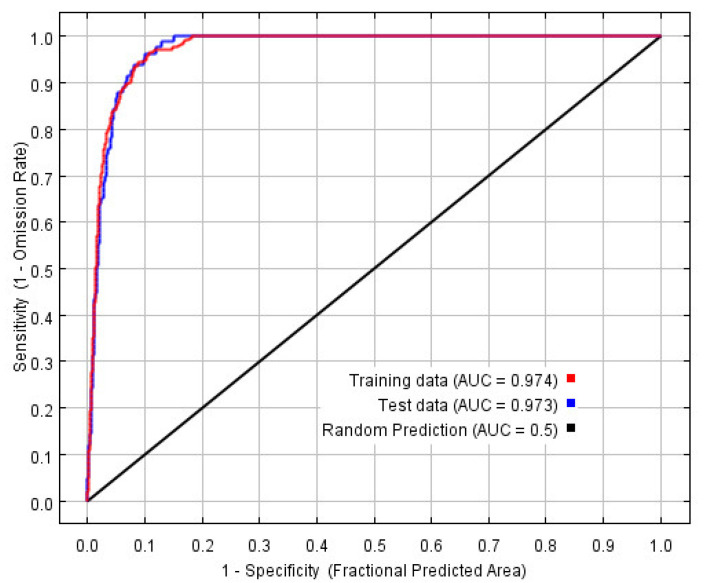
Receiver operating characteristic curve of *S. invicta*.

**Figure 4 insects-12-00874-f004:**
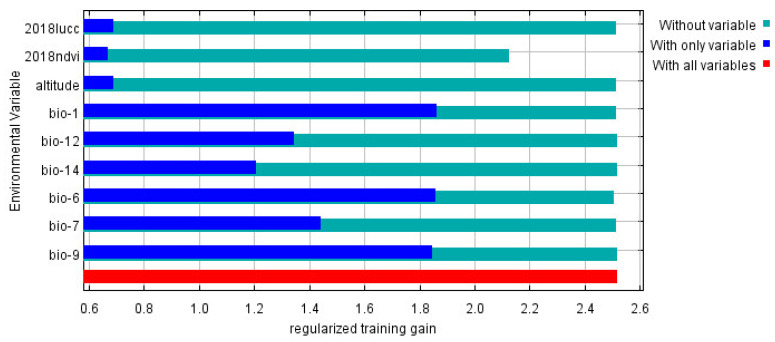
Jackknife test result of environmental factor for *S. invicta*.

**Figure 5 insects-12-00874-f005:**
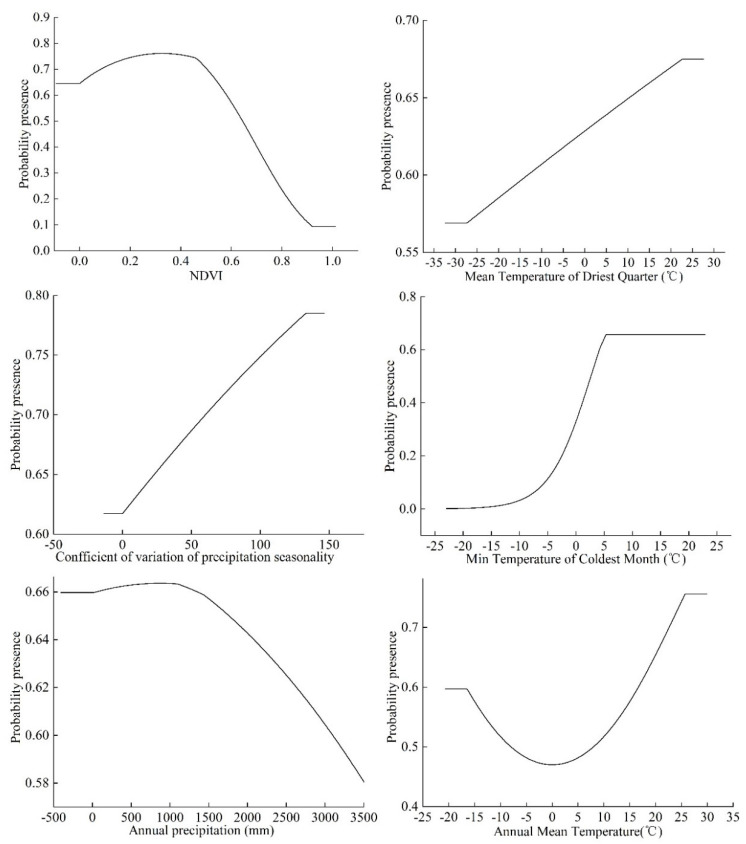
Response curves of probability of the presence of *S. invicta*.

**Figure 6 insects-12-00874-f006:**
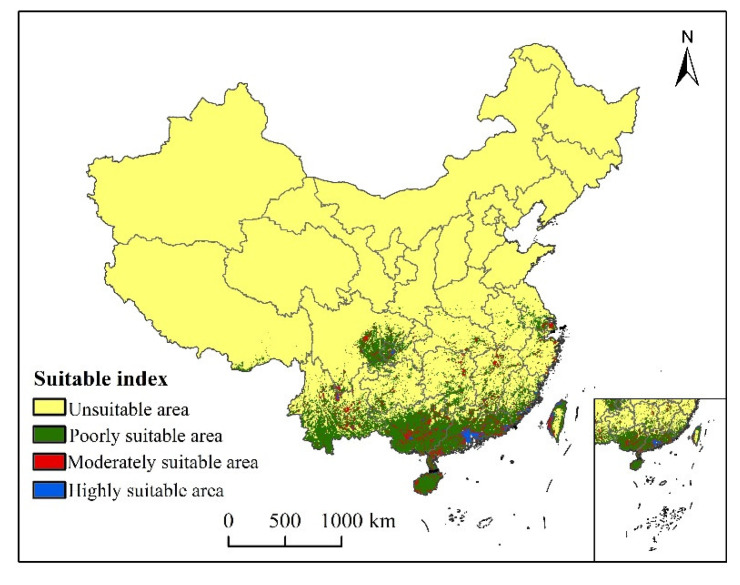
Potential current and suitable area of *S. invicta* growth in China.

**Figure 7 insects-12-00874-f007:**
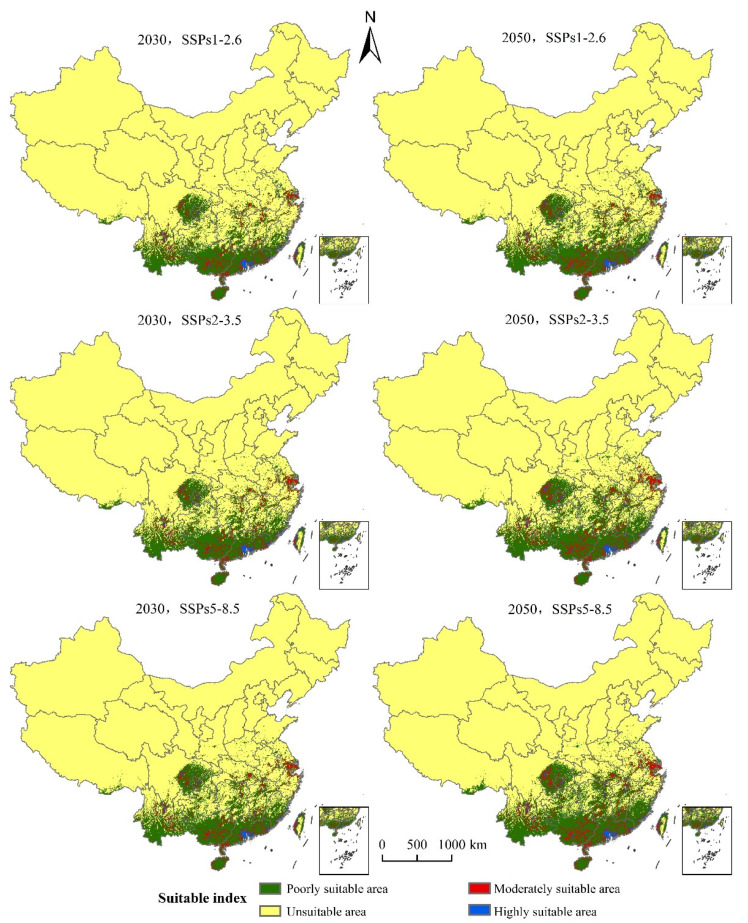
Potentially suitable area of *S. invicta* growth under different climate change scenarios in China.

**Figure 8 insects-12-00874-f008:**
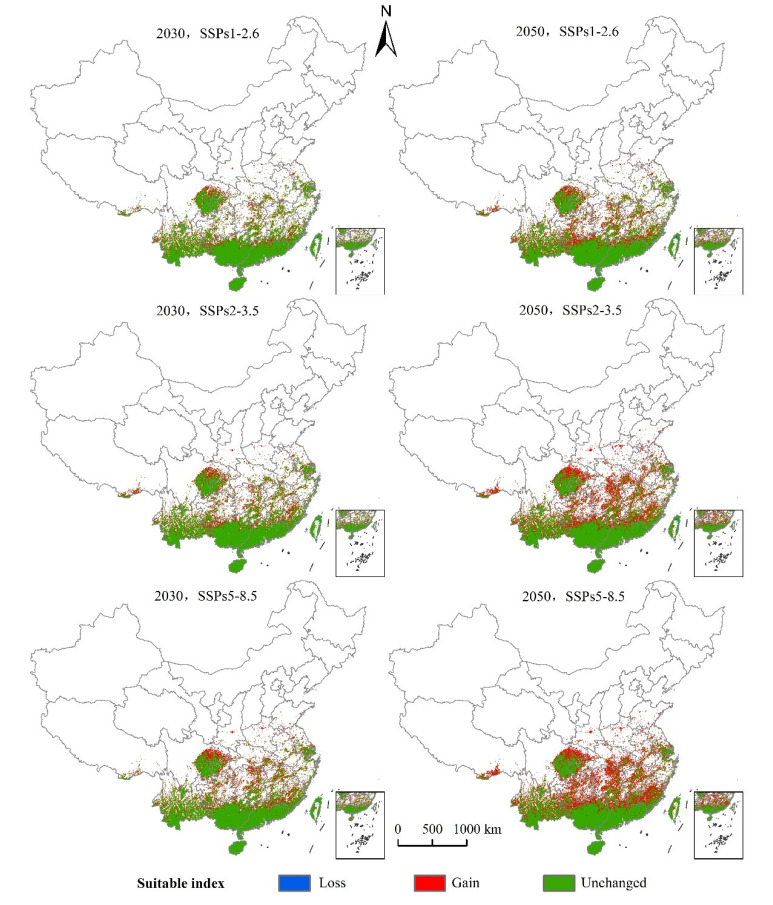
Changes in the potential geographical distribution of *S. invicta* under climate change scenarios.

**Figure 9 insects-12-00874-f009:**
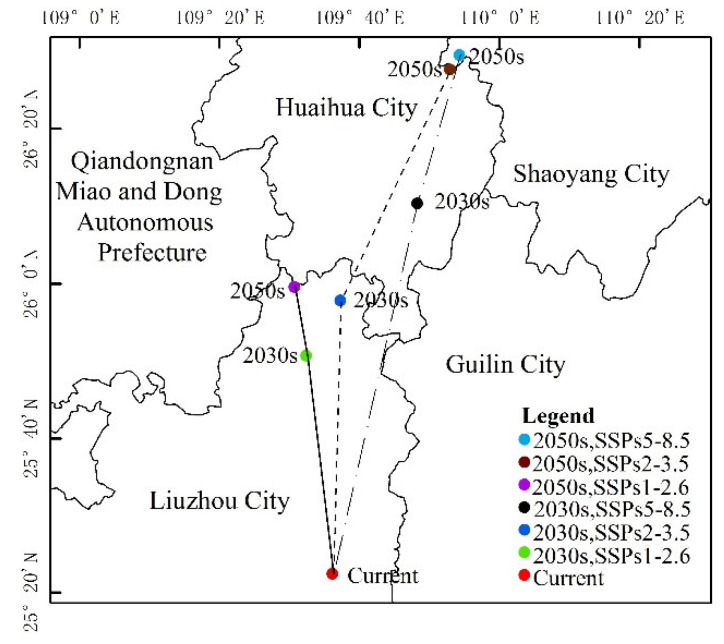
Highly suitable area centroid distributional shifts under climate change for *S. invicta*.

**Table 1 insects-12-00874-t001:** Environmental factor data used in this study and the final screening modeling data.

Code	Description	Whether to Use for Modeling
Bio1	Annual Mean Temperature (°C)	Yes
Bio2	Mean Diurnal Range (Mean of monthly (max temp–min temp)) (°C)	No
Bio3	Isothermality (BIO2/BIO7) (×100)	No
Bio4	Temperature Seasonality (standard deviation ×100)	No
Bio5	Max Temperature of Warmest Month (°C)	No
Bio6	Min Temperature of Coldest Month (°C)	Yes
Bio7	Temperature Annual Range (BIO5-BIO6) (°C)	Yes
Bio8	Mean Temperature of Wettest Quarter (°C)	No
Bio9	Mean Temperature of Driest Quarter (°C)	Yes
Bio10	Mean Temperature of Warmest Quarter (°C)	No
Bio11	Mean Temperature of Coldest Quarter (°C)	No
Bio12	Annual precipitation (mm)	Yes
Bio13	Precipitation of Wettest Month (mm)	No
Bio14	Coefficient of Variation of Precipitation Seasonality	Yes
Bio15	Precipitation Seasonality (Coefficient of Variation)	No
Bio16	Precipitation of Wettest Quarter (mm)	No
Bio17	Precipitation of Driest Quarter (mm)	No
Bio18	Precipitation of Warmest Quarter (mm)	No
Bio19	Precipitation of Coldest Quarter (mm)	No
NDVI	Normalized Vegetation Index	Yes
Altitude	Altitude (m)	Yes
Slope	Slope (°)	No
Aspect	Aspect	No
LUCC	Land Use and Land Cover Change	Yes

**Table 2 insects-12-00874-t002:** Suitable areas of *S. invicta* growth under different climate change scenarios (10^4^ km^2^).

Period	Highly Suitable	Moderately Suitable	Poorly Suitable	Total Suitable
Current	3.24	10.66	67.47	81.37
2030s, SSPs1-2.6	3.87	14.44	84.54	98.98
2030s, SSPs2-4.5	4.00	15.40	88.78	108.18
2030s, SSPs5-8.5	4.26	16.05	92.83	113.14
2050s, SSPs1-2.6	4.29	16.31	92.62	113.22
2050s, SSPs2-4.5	5.02	19.42	104.68	129.12
2050s, SSPs5-8.5	5.46	20.94	110.31	136.71

**Table 3 insects-12-00874-t003:** Future changes in suitable habitat area (10^4^ km^2^).

Period	Loss	Gain	Unchanged
2030s, SSPs1-2.6	0.08	19.32	72.96
2030s, SSPs2-3.5	0.04	23.89	73.00
2030s, SSPs5-8.5	0.05	28.60	72.99
2050s, SSPs1-2.6	0.14	28.42	72.89
2050s, SSPs2-3.5	0.11	42.91	72.93
2050s, SSPs5-8.5	0.01	49.44	73.03

## Data Availability

The data is included in the article. For the data provided in this study, see the Section 2.2 and Section 2.3 in the text.

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
