# Peer review of "Prediction of Spatiotemporal Invasive Risk of the Red Import Fire Ant, Solenopsis invicta (Hymenoptera: Formicidae), in China"

_insects, 2021, doi:10.3390/insects12100874_

Round 1
Reviewer 1 Report
Dear authors
Thanks for this interesting study and the opportunity to evaluate your study. I have some concerns, especially regarding the discussion of your results. I am missing the relation of the ecology of the species to the specific environmental variables. Please elaborate on the ecology of S. invicta in the material and methods section.
Based on the ecology you should try to explain how the environmental variables which proved to be important are affecting the distribution of S. invicta. The discussion has to be improved and needs elaboration on the specific environmental variables, e.g. Bio1, Bio6, Bio9...Additionally I am missing information on the mobility and dispersal characteristics of S. invicta. A suitable habitat alone doesn't necessarily mean that the species is capable of reaching it.

Author Response
Dear Editors and Reviewers:
Thank you for your letter and for the reviewers’ comments concerning our manuscript entitled “Prediction of Spatiotemporal Invasive Risk by the Red Import Fire Ant, Solenopsis invicta (Hymenoptera: Formicidae) in China” (ID:insects-1353851 ). Those comments are all valuable and very helpful for revising and improving our paper, as well as the important guiding significance to our researches. We have studied comments carefully and have made correction which we hope meet with approval. Revised porion are marked up the“Track Changes”function in the paper.
The manuscript has been edited by MDPI English service.
The main corrections in the paper and the responds to the reviewer’s comments are as flowing:
Responds to the reviewer’s comments:
- Thanks for this interesting study and the opportunity to evaluate your study. I have some concerns, especially regarding the discussion of your results. I am missing the relation of the ecology of the species to the specific environmental variables. Please elaborate on the ecology of invicta in the material and methods section. Based on the ecology you should try to explain how the environmental variables which proved to be important are affecting the distribution of S. invicta. The discussion has to be improved and needs elaboration on the specific environmental variables, e.g. Bio1, Bio6, Bio9...Additionally I am missing information on the mobility and dispersal characteristics of S. invicta. A suitable habitat alone doesn't necessarily mean that the species is capable of reaching it.
Response: Thank you for your suggestion. I have supplemented the ecological situation of S.invicta in detail in the Material and Method section, and improved this part of the material and method according to the suggestion that you annotated in the manuscript. According to your suggestion, I have carefully revised the discussion section by adding the influence of specific environmental variables on the existence probability of S.invicta, and supplemented the information of S.invicta mobility and diffusion, and given preventive suggestions combined with the results of this study. The conclusion section has also been revised accordingly. Other modifications have also been made in detail according to the suggestions that you annotated in the manuscript. The specific modifications have been marked in the revised manuscript with the function of “Track Change”.
- The questions that need to be explained and answered in the annotations of manuscript are as follows:
1、In Figure 1:Please add some additional information, e.g. What is shown in this figure? Where did you get the data?
Response 1:Thank you for your problem and suggestion. Figure 1 shows the distribution data of S.invicta in China, and the data acquisition was described in detail in the first paragraph of 2.2 section.
- 2、About “survival probability” in the second paragraph of 3.2 section:Maybe I am overlooking something, but how are you estimating the survival probability?
- You are modeling the presence of the species which says, in my opinion very little about the survival probability. Can you please clarify and elaborate on this point.
Response 2:Thank you for your problem and suggestion. I am sorry for the inconvenience caused by our inappropriate expression. According to the suggestion of another reviewer; I have changed "survival probability" to "probability of presence".
The probability presence of S.invicta predicted by MaxEnt model is continuous raster data, and the value is between 0 and 1, the closer the value is to 1, the higher the possibility of species existence. Accordingly, the suitable zone of S.invicta is divided into four grades according to their probability presence:unsuitable area<0.1; 0.1≤poorly suitable area<0.3; 0.3≤moderately suitable area<0.6; and highly suitable area≥0.6.
The subject of this paper is to investigate how the potential geographic distribution of S.invicta changes in the context of climate change using the MaxEnt model, and to investigate how the probability presence of S.invicta changes with changes in environmental variable factors, so as to obtain the suitable range of each environmental variable factor in terms of changes in the probability presence of S.invicta. The model results are represented by the probability of species occurrence, with higher values indicating that environmental conditions are more suitable for species survival.
Reviewer 2 Report
The red fire ant is an important invasive species, and it is expected that in the future, its impact will increase. The topic is therefore of relevance to the journal. The application is appropriate, and the results are useful. The general presentation is, while understandable, necessary to be improved. I made numerous linguistic suggestions.
Simple summary: this is not really simple - it is nearly identicla to the subsequent "expert summary". You'd need to greatly simplify this part.
line 25 - only species but not higher categories need Italics.
62-64 We cannot prevent the harm caused by invasive species, no matter what we try. You could, at maximum, reduce the damage caused by them. Rephrase this part.
81-85 these statement need citations - the species is well studied, so all these statements can be supported by published evidence.
94 "the second batch of alien invasive species" - what does this mean? This is not clear
Be aware that your results will have to be presented using the past tense - you seem to consistently use the pressent tense. This is wrong.
Fig. 1: delete the bottom panel from the map - not relevant for this work. Delete this irrelevant detail also from the other maps where present.
Table 1 - these are known scenarios, to there is no need to have this table. Refer to them only by their citation.
135 6th International Coupled Modes Comparison Plan
137 you seem to be confused about the nature of climatic data vs. biological data. Biological data are generated by living organisms - what you have on table 2 are climatic data (except NVDI)
150-ff what you describe here is routine in such investigations. No need to go to this level of detail.
part 2.3. - it looks like you have recently learned how to use the model and decided that you would describe everything you did. This is not necessary. You have only to mention what method you used, and only mention details that were not routine. If readers want to know the method better, tey will not read this paper - they go to the instructions of the model.
Point 2.4. - the same here. You do not have to replicate what is in the instructions manual. Both of these need a complete re-writing.
Fig 2 - my understanding is that you used the recorded fire ant occurrences in China. The resulting analysis is therefore not the "native niche" model. For that, you'd need data from the native area. Modify the text under fig. 2. accordingly.
230-231 I suggest that you rename the variables - it is misleading to call climatic variables "Bio" - these are, as I mentioned above, not biological variables.
Fig 4 is redundant: all information is already mentioned in the text. Delete.
paragraph starting on 238: this is a confused and confusing description. You do not have to give information how to interpet model outcomes. Only concentrate on the results. There are also misinterpretations, such as: "the corresponding environmental factor value is the threshold of S.invicta beginning to exist" - no, the factor exists independently of the presence of S.invicta - it is then considered influential. This is a substantially different thing.
254 'survival probability' - the term is suspect. You probably hav eno data on the actual survival of the species at given locations. You have occurrence data, and from this, it is risky to draw conclusiions abotu the species' survival at that location. In fact, the vertical axes are "probability of presence". That is a more fortunate term.
269 land are of China? Modify
268-71 vs. table 3: this is double data presentation. No need for this, keep the numbers on the table only - simplify the text.
280-ff: I suggest that you detail the outcome of various scenarios in different paragraphs. It is difficult to follow as it is now.
357 you have to include references to support this statement
384 what do you mean by "species vegetative pressure"?
417-418 I find these suggestions unachievable. You can reduce spread but will not be able to prevent it. Rephrase.

Author Response
Dear Editors and Reviewers:
Thank you for your letter and for the reviewers’ comments concerning our manuscript entitled “Prediction of Spatiotemporal Invasive Risk by the Red Import Fire Ant, Solenopsis invicta (Hymenoptera: Formicidae) in China” (ID:insects-1353851 ). Those comments are all valuable and very helpful for revising and improving our paper, as well as the important guiding significance to our researches. We have studied comments carefully and have made correction which we hope meet with approval. Revised porion are marked up the“Track Changes”function in the paper.
The manuscript has been edited by MDPI English service.
The main corrections in the paper and the responds to the reviewer’s comments are as flowing:
Responds to the reviewer’s comments:
- Simple summary: this is not really simple - it is nearly identicla to the subsequent "expert summary". You'd need to greatly simplify this part.
Response: Thank you for your suggestion. I have simplified the content of Simple summary in lines 10-17age 1 of the revised manuscript.
- line 25 - only species but not higher categories need Italics.
Response: Thanks for your suggestion’, I have changed "Solenopsis invicta (Hymenoptera: Formicidae)" to "Solenopsis invicta (Hymenoptera: Formicidae)".
- 62-64 We cannot prevent the harm caused by invasive species, no matter what we try. You could, at maximum, reduce the damage caused by them. Rephrase this part.
Response: Thank you for your suggestion, I have restated the content of this part in lines 58-61 of page 2 of the revised manuscript.
- 81-85 these statement need citations - the species is well studied, so all these statements can be supported by published evidence.
Response: Thank you for your suggestion, I have quoted references to the content of this part, so that these claims can be supported by published evidence.
- 94 "the second batch of alien invasive species" - what does this mean? This is not clear
Response: Thank you for your problem and suggestion. I am sorry for the inconvenience caused by our inappropriate expression. The S.invicta has a mixed feeding habit, strong competitiveness, and has obvious harm to the natural environment, human health, agriculture and forestry, etc. Therefore, it was listed as the second batch of alien invasive species in China by the Ministry of Ecology and Environment of the people's Republic of China in 2010. In addition, I quote references to support this claim. See paragraphs 2 of the introduction on page 65-68 for specific revisions.
- Be aware that your results will have to be presented using the past tense - you seem to consistently use the pressent tense. This is wrong.
Response: We are very sorry for the inconvenience caused by the language of the article. Therefore, we have asked the professional language editing company (MDPI) to polish the language of the article.
- 1: delete the bottom panel from the map - not relevant for this work. Delete this irrelevant detail also from the other maps where present.
Response: Thank you for your suggestion. I have modified the Figure 1 See Figure 1 on page 4 for the revision.
- Table 1 - these are known scenarios, to there is no need to have this table. Refer to them only by their citation.
Response: Thank you for your suggestion, I have deleted the table 1 in original manuscript.
- 135 6th International Coupled Modes Comparison Plan
Response: Thank you for your suggestion, I have revised the content of this part.
- 137 you seem to be confused about the nature of climatic data vs. biological data. Biological data are generated by living organisms - what you have on table 2 are climatic data (except NVDI)
Response: Thank you for your suggestion. This is where I made a mistake. I checked the introduction of variable data used in this study (http://www.worldclim.org/), and the name should be "Bioclimatic Variables". These variables are introduced in detail by Booth et al.
- 150-ff what you describe here is routine in such investigations. No need to go to this level of detail.
Response: Thank you for your suggestion, I have deleted the content of this part.
- part 2.3. - it looks like you have recently learned how to use the model and decided that you would describe everything you did. This is not necessary. You have only to mention what method you used, and only mention details that were not routine. If readers want to know the method better, tey will not read this paper - they go to the instructions of the model.
Response: Thanks for your suggestion, combined with the suggestion of another reviewer, I have revised this part of the content. See the 2.4 section on page 5 for specific revisions.
- Point 2.4. - the same here. You do not have to replicate what is in the instructions manual. Both of these need a complete re-writing.
Response: Thank you for your suggestion. See the 2.5 section on page 5-6 for specific revisions. The newly revised content can be found in heading 2.5.
- Fig 2 - my understanding is that you used the recorded fire ant occurrences in China. The resulting analysis is therefore not the "native niche" model. For that, you'd need data from the native area. Modify the text under fig. 2. accordingly.
Response: Thanks for your suggestion, I have changed the title of Figure 2 on page 6.
- 230-231 I suggest that you rename the variables - it is misleading to call climatic variables "Bio" - these are, as I mentioned above, not biological variables.
Response: Thank you for your suggestion, I consulted the introduction of variable data used in this study (http://www.worldclim.org/), and the name should be "Bioclimatic Variables", abbreviated as "Bio".
- Fig 4 is redundant: all information is already mentioned in the text. Delete.
Response: Thank you for your suggestion, I have deleted the Figure 4 in original manuscript.
- paragraph starting on 238: this is a confused and confusing description. You do not have to give information how to interpet model outcomes. Only concentrate on the results. There are also misinterpretations, such as: "the corresponding environmental factor value is the threshold of S.invicta beginning to exist" - no, the factor exists independently of the presence of S.invicta - it is then considered influential. This is a substantially different thing.
Response: Thank you for your suggestion, I have deleted this part of the content and only retained the results.
- 254 'survival probability' - the term is suspect. You probably hav eno data on the actual survival of the species at given locations. You have occurrence data, and from this, it is risky to draw conclusiions abotu the species' survival at that location. In fact, the vertical axes are "probability of presence". That is a more fortunate term.
Response: Thank you for your suggestion. I have changed the "survival probability" to "probability of presence" in full text.
- 269 land are of China? Modify
Response: Thank you for your suggestion. The projection error in Figure 7 in original manuscript. See Figure 6 on page 9 for the revision.
- 268-71 vs. table 3: this is double data presentation. No need for this, keep the numbers on the table only - simplify the text.
Response: Thank you for your suggestion, I have simplified the content of this part in lines 233-235 of page 9 of the revised manuscript.
- 280-ff: I suggest that you detail the outcome of various scenarios in different paragraphs. It is difficult to follow as it is now.
Response: Thank you for your suggestion. I have divided the two paragraphs of heading 3.4 into small paragraphs.
- 357 you have to include references to support this statement
Response: Thank you for your suggestion, in combination with the suggestion of another reviewer, we revised the discussion section and added references where citations are needed.
- 384 what do you mean by "species vegetative pressure"?
Response: Thank you for your suggestion, this is where I chose the wrong words, which has been revised to "Species propagule pressure". Species propagule pressure is a comprehensive expression of the number of individual organisms released to non-native areas, which is a combination of the number of reproductive forms, released each time and the number of times released. Propagule pressure is an important condition for the success of alien species invasion.
- 417-418 I find these suggestions unachievable. You can reduce spread but will not be able to prevent it. Rephrase.
Response: Thank you for your suggestion, combined with the suggestion of another reviewer and according to the results of the article to simplify the content of this part.
References:
Booth T H, Nix H A, Busby J R, et al. bioclim: the first species distribution modelling package, its early applications and relevance to most current MaxEnt studies[J]. Diversity & Distributions, 2013, 20(1):1-9.
Round 2
Reviewer 1 Report
Dear authors
Thank you for this revised version of the manuscript. You basically addressed my major comments and concerns regarding the manuscript.
All the best